# DeepOmni: Towards Seamless and Smart Speech Interaction with Adaptive Modality-Specific MoE

## Abstract

Native multimodal large language models (MLLMs) restructure a single large language model (LLM) into a spoken language model (SLM) capable of both speech and text generation. Compared to modular and aligned MLLMs, native MLLMs preserve richer paralinguistic features such as emotion and prosody, and generate speech responses directly within the backbone LLM rather than using a separate speech decoder. This integration also results in lower response latency and smoother interaction. However, native MLLMs suffer from catastrophic forgetting and performance degradation because the available paired speech-text data is insufficient to support the pretraining of MLLMs compared to the vast amount of text data required to pretrain text LLMs. To address this issue, we propose DeepOmni, a framework for adaptive modality expert learning based on a Mixture of Experts (MoE) architecture. DeepOmni first adaptively distinguishes modality experts according to their modality load within the LLM. Each modality expert then undergoes specialized single-modality training, followed by joint multimodal collaborative training. As a result, DeepOmni incurs only a $5.5\%$ performance drop compared to the original LLM, which is significantly lower than the average performance drop of over $20\%$ typically seen in native MLLMs (such as GLM-4-Voice), and is on par with modular MLLMs. Meanwhile, the end-to-end dialogue latency remains within $0.5$ seconds, ensuring a seamless and intelligent speech interaction experience.

## 1 Introduction

Since GPT-4o OpenAI (2024) has demonstrated the great potential of using a unified model to process speech, recent voice interaction systems have evolved from traditional cascaded systems to end-to-end large speech interaction models. As shown in Figure 1 a), in traditional cascaded systems input speech is first transcribed into text by an automatic speech recognition (ASR) Radford et al. (2023); Bai et al. (2024); Povey et al. (2011), the resulting text is then comprehended by a large language model (LLM) Liu et al. (2024); Yang et al. (2024); GLM et al. (2024) to generate a text response, and finally the response is synthesized into speech by a text-to-speech (TTS) Du et al. (2024); Ren et al. (2019); Wang et al. (2025) model. However, this cascaded structure results in high latency, and errors from each module can accumulate and propagate, leading to degraded performance. Therefore, recent speech LLMs adopt an end-to-end approach, using a unified model to process both speech and text simultaneously, which enhances the capabilities of speech understanding and generation. At the same time, this approach significantly reduces interaction latency and provides a more natural and smooth voice interaction experience.

Current speech LLMs can be divided into two main categories: modular aligned multimodal and native multimodal Chen et al. (2025). Representatives of the former include Qwen2.5-Omni Xu et al. (2025), Minmo Chen et al. (2025), LLaMA-Omni Fang et al. (2024), Freeze-Omni Wang et al. (2024), and VITA-1.5 Fu et al. (2025). These models connect a speech encoder and a speech decoder to the LLM to handle audio understanding and audio generation tasks separately, placing semantic understanding within the LLM module. This maximally the retains LLM's general capabilities. Furthermore, since the LLM itself does not require retraining, much less speech-text paired pretraining data is needed.

Representative of the latter including Mini-Omni Xie & Wu (2024), GLM-4-Voice Zeng et al. (2024), Moshi Défossez et al. (2024), LUCY Gao et al. (2025) and Step-Audio2 Wu et al. (2025) use large amounts of audio-text paired data to retrain the LLM into a spoken language model, enabling simultaneous output of both audio and text tokens. Compared to modular aligned multimodal approaches, native ones greatly reduces the latency of speech token output. More paralinguistic information in speech, such as emotion and prosody, is retained within the SLM rather than in the speech decoder, benefiting the full expression of paralinguistic information. However, since native multimodal models expand the original LLM's vocabulary and require retraining, they need a large amount of audio-text paired pretraining data. Therefore, in real-world scenarios where audio data is much less abundant than text data, native multimodal models are prone to catastrophic forgetting, severely impairing the original LLM's language capabilities.

In this paper, to alleviate the catastrophic forgetting faced by native multimodal models Zhai et al. (2023), we propose DeepOmni, a model that applies the mixture of experts (MoE) for multimodal learning Li et al. (2025b; 2024); Zhong et al. (2024); Shen et al. (2024) to effectively isolate modality-specific knowledge. Since most speech dialogue data are conversational and colloquial, while text LLMs are primarily trained on formal written text, training LLMs with colloquial speech dialogue data can lead to less standardized written outputs, resulting in decreased language performance on certain LLM benchmarks. Therefore, we assign different parameters within the LLM to specialize in instruction data from different modalities, allowing modality experts to focus on single-modality training and avoid interference between modalities. Finally, we use cross-modal instruction data to jointly train the modality experts, enabling the model to output both speech and text simultaneously.

To minimize the damage to the original text LLM, we propose an adaptive modality expert selection strategy, which dynamically selects modality experts based on the MoE model's modality load on different data. Experts with a high speech token load but a low text token load are selected as audio experts. After several iterations of modality expert selection and training, DeepOmni achieves only a loss of $5.5\%$ relative in language ability, reaching the same level of performance loss as modular multimodal models.

In summary, the contributions of this paper are as follows:

• We propose DeepOmni, which designates modality experts within the MoE to separate interference between modalities and isolate modality-specific knowledge. At the same time, modalities can collaborate to jointly output multimodal results, enabling both colloquial and formal language to coexist within a single model.

• We introduce an adaptive modality expert selection method based on modality load, which further reduces the damage to the original text LLM and alleviates the problem of catastrophic forgetting.

• To the best of our knowledge, this is the first native multimodal large speech interaction model based on the MoE architecture. It opens up a new direction for mitigating catastrophic forgetting in native multimodal speech interaction models and bridges the gap between native multimodal and modular multimodal approaches.

## 2 RELATED WORK

### 2.1 END-TO-END SPEECH INTERACTION SYSTEM

Speech is among the most important signal in human communication, as it not only conveys semantic content but also carries various paralinguistic information such as emotion, speech rate, intonation, and timbre. As a result, speech interaction has become an increasingly important mode of human-computer interaction. Traditional speech interaction systems are typically composed of a pipeline of separately optimized ASR, LLM, and TTS modules. Such systems not only suffer from high interaction latency, but also accumulate errors from each module, leading to a less satisfactory overall user experience. Therefore, end-to-end solutions that use a unified model to process both speech and text simultaneously have become the mainstream approach Ji et al. (2024a).

Some of the early attempts integrate ASR into LLM by connecting the ASR encoder directly to the LLM via an adapter, rather than directly feeding the ASR transcript into the LLM Kong et al. (2020); Chu et al. (2024; 2023); Das et al. (2024). This approach not only provides the LLM with richer speech information than using pure text input, but also reduces the latency caused by ASR recognition. However, these models still require an additional TTS module to achieve end-to-end

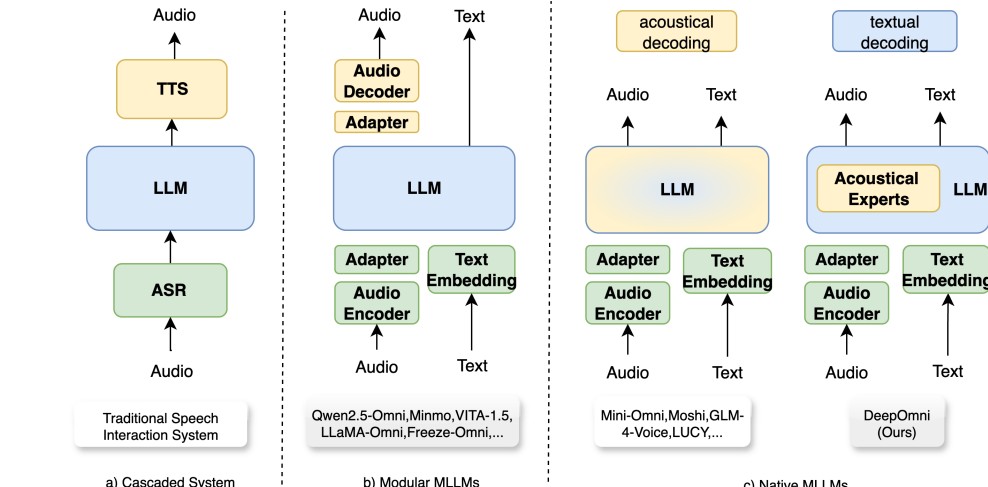

Figure 1: Comparison of DeepOmni and existing voice interaction systems. Compared to modular MLLMs, DeepOmni embeds acoustic experts into a single pre-trained LLM and integrates both speech and language decoding within a single LLM. Through adaptive modality expert selection and a three-stage training process for modality experts, DeepOmni effectively alleviates catastrophic forgetting in text-based LLMs while maintaining robust voice interaction capabilities.

speech interaction. To provide rich speech information, LLaMA-Omni Fang et al. (2024) feed LLM's text hidden states to a non-autoregressive (NAR) Transformer based TTS module to predict discrete speech tokens. Since speech token sequences are much longer than text sequences for the same content, the text hidden states are upsampled and a Connectionist Temporal Classification (CTC) loss Graves et al. (2006) is used to align the upsampled hidden states with the audio token outputs. Meanwhile, to reduce latency for real-time speech interaction, a predefined chunk size is set to enable the vocoder to synthesize speech in a streaming manner. Mini-Omni employs SNAC, an audio codec, to discretize continuous acoustic signal into 7 codebooks of discrete tokens at a rate of 82Hz and adopts a delayed decoding strategy Copet et al. (2023); Peng et al. (2024); Lyth & King (2024) to model 7 streams of acoustic tokens to effectively reduces the number of steps of LLM decoding during inference. Minmo Chen et al. (2025) integrates a language model with an autoregressive CosyVoice2 decoder Du et al. (2024). The hidden states of the language model are fed into the CosyVoice2 decoder to autoregressively predict speech tokens, which are then passed to a vocoder for waveform synthesis.

Fully end-to-end approaches integrate the TTS module into the LLM, allowing the LLM itself to generate both text and speech tokens, rather than relying on a separate speech decoder or TTS module for speech synthesis. Discrete speech tokens are typically generated by neural audio codecs Défossez et al. (2022); Ji et al. (2024b); Zhang et al. (2023) or self-supervised models Lakhotia et al. (2021); Hsu et al. (2021); Chen et al. (2022). Currently, there are two mainstream modeling paradigm for end-to-end speech systems. The first is interleaved audio-text modeling, as exemplified by GLM-4-Voice Zeng et al. (2024), Baichuan-Audio Li et al. (2025a) and Spirit-LM Nguyen et al. (2025), where audio and text tokens are interleaved within a single sequence, and the model alternately predicts audio and text tokens. The second is parallel audio-text modeling, represented by models such as Mini-Omni Xie & Wu (2024), Moshi Défossez et al. (2024), Slam-Omni Chen et al. (2024) and LUCY Gao et al. (2025). In this paradigm, text and audio tokens can be generated in parallel: while generating text tokens, multiple LM heads are used to simultaneously predict multiple audio tokens. Compared to the interleaved approach, parallel modeling compresses speech tokens into shorter sequences, enabling the modeling of higher-bitrate speech tokens.

**Modular Speech Language Models** Modular large speech models are primarily built on an architecture where a LLM is directly connected to an audio encoder and decoder through adapters. Examples include Qwen2.5-Omni Xu et al. (2025), Minmo Chen et al. (2025), LLaMA-Omni Fang et al. (2024), and Freeze-Omni Wang et al. (2024), as shown in Fig. 1 b). In this structure, the LLM is only responsible for textual decoding, while speech decoding is handled by the audio decoder. As a result, the language capabilities of the LLM are largely preserved. Additionally, Minmo employs

Low-Rank Adaptation (LoRA) Hu et al. (2022) for fine-tuning the LLM, while Freeze-Omni further freezes the LLM to prevent any degradation of its language abilities. However, such modular speech-language models face several challenges. For instance, the transmission of speech information relies entirely on the audio decoder, with the LLM only responsible for conveying textual content. This means that the expression of paralinguistic information in speech is completely dependent on the speech decoder, resulting in underutilization of the LLM. Furthermore, there are additional issues such as low deployment efficiency. In modular speech language models paradigm, the LLM is trained to model only text distribution, or formally $P(T_{out}|T_{in} \cup S_{in})$, and the audio decoder is trained to model $P(S_{out}|T_{out})$.

**Native Speech Language Models** Native speech language models reconstruct the LLM into an SLM, enabling the LLM to output both text and speech tokens simultaneously, as shown in Fig. 1 c). Currently, most mainstream native multimodal approaches such as Mini-Omni Xie & Wu (2024), Moshi Défossez et al. (2024), GLM-4-Voice Zeng et al. (2024), and LUCY Gao et al. (2025) have the LLM handle both text and speech decoding. This approach offers low latency, easy deployment, and allows a unified model to process both audio and text decoding, which better aligns with end-to-end requirements. However, without large-scale speech-text paired data, retraining the LLM in this way can lead to catastrophic forgetting and performance degradation. In native speech language models paradigm, the LLM is trained to model $P(T_{out} \cup S_{out}|T_{in} \cup S_{in})$ . Clearly in the native paradigm, the speech-text joint distribution space for the LLM to learn is more complex than text distribution. It is also for this reason in modular paradigm the LLM does not need to be restructed to a SLM and its language capacity is largely preserved.

## 2.2 MULTIMODAL MIXTURE-OF-EXPERTS

Multimodal Mixture-of-Experts refers to using a MoE model to learn multimodal knowledge. BEiT-3 Wang et al. (2023), VLMo Bao et al. (2022), and Uni-MoE Li et al. (2025b) employ specific modality expert groups to capture modality-specific information. MoExtend Zhong et al. (2024) further expands modality experts horizontally on top of existing LLMs to capture additional modality information. CuMo Li et al. (2024) extends dense models into multimodal LLMs by co-upcycling the MLP, while MoME Shen et al. (2024) uses multiple feature encoders as modality expert groups to encode multi-dimensional features. VL-MoE Shen et al. (2023) and MoE-LLaVa Lin et al. (2024a) introduce mixture-of-experts (MoE) to improve training and deployment. MoMa Lin et al. (2024b) pretrains MLLMs with multimodal mixture-of-experts and collaborates with sparse components to enhance the efficiency of training from scratch with trillions of mixed-modality tokens. Inspired by these works, we introduce multimodal mixture-of-experts (speech experts and text experts) into end-to-end large speech interaction models for native MLLM pretraining. At the same time, we use the modality load of each expert to adaptively select modality experts, a method called Adaptive Modality-Specific MoE, to address the catastrophic forgetting problem aforementioned.

## 3 METHODS

### 3.1 ARCHITECTURE

The overall architecture of the model is shown in Fig. 2, which mainly consists of an Audio Encoder, a connector, an MoE backbone, and a streaming codec. Let $X_i^A$ and $X_i^T$ denote the audio input and text input of the $i$-th sample, respectively, and $Y_i^A$ and $Y_i^T$ denote the audio output and text output of the $i$-th sample, respectively. The model's text and speech responses are represented as $Y^T \in V$ and $Y^A \in U$, where $V$ and $U$ denote the text vocabulary and the speech codec vocabulary, respectively. The embeddings of speech $H_i^A$ and text $H_i^T$ are combined and used as the overall feature $H_i$ input to the model. The loss $L$ of the model over N samples can be defined as:

$$L = -\sum_{i=1}^{N}\sum_{t=1}^{T_i} logP(Y_{i,t}^T, Y_{i,t}^A|Y_{i,<t}^T, Y_{i,<t}^A, H_i), \tag{1}$$

where $T_i$ denotes the maximum number of tokens of the output text $Y_i^T$ and the output speech $Y_i^A$ of the $i$-th sample.

**Audio Encoder and Adapter** We use Whisper-medium Radford et al. (2023) as the audio encoder, which consists of 24 Transformer blocks with a hidden size of 4096 and 16 attention heads. It includes

multiple convolutional downsampling layers that downsample the speech features by a factor of 4. We use an 80-dimensional log Mel-filterbank with a 25ms window length computed every 10ms as the input to the audio encoder. A MLP-based audio adapter is used to align the audio modality with the backbone language model and further downsample the speech features. Finally, for the input $X_i^A$, we obtain the hidden state $H_i^A = Adapter(Encoder(X_i^A))$.

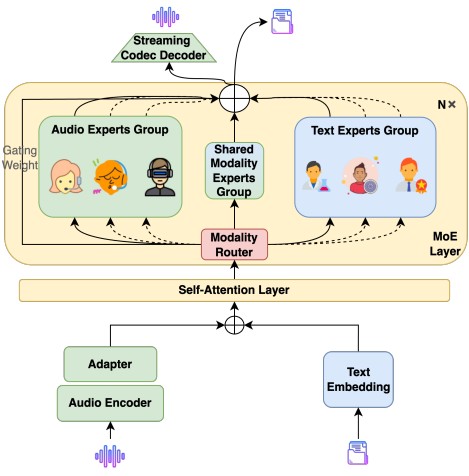

Figure 2: The DeepOmni model architecture.

Table 1: Performance of Text to Speech on Seed-TTS. Lower values are better. The metric for Chinese is character error rate (CER), and the metric for English is word error rate (WER).

| Model | Seed-TTS | | |
| --- | --- | --- | --- |
| | test-zh↓ | test-en↓ | test-hard↓ |
| Seed-TTS | 1.12 | 2.25 | 7.59 |
| CosyVoice | 3.63 | 4.29 | 11.75 |
| CosyVoice2 | 1.45 | 2.57 | 6.83 |
| GLM-4-Voice | 2.91 | 2.10 | - |
| VITA-1.5 | 8.44 | 2.63 | - |
| DeepOmni(topk=10) | 1.87 | 7.34 | 7.53 |
| DeepOmni(topk=1) | 1.68 | 6.28 | 7.34 |
| +DPO | 1.41 | 3.25 | 7.29 |

**Mixture-of-Experts Backbone** The backbone model is DeepSeek-V2-Lite Liu et al. (2024), as shown in Fig. 2. Each MoE layer is adaptively divided into modality expert groups for specialized training and learning of the corresponding modality. Specifically, the text expert group is responsible for generating written text such as mathematical code, while the speech expert group is responsible for generating spoken language such as emotional speech response. In addition, there is a shared modality expert group for learning common knowledge across both modalities such as daily conversations. The weights among modality expert groups are allocated through modality routing, as follows:

$$P(h_l)_i = \frac{e^{f(h_l)_i}}{\sum_j^M e^{f(h_l)_{i,j}}}, \tag{2}$$

$$MoE(h_l) = \sum_{i=1}^T P(h_l)_i \cdot e(h_l)_i, \tag{3}$$

where $h_l$ denotes the averaged feature input of speech and text at the $l$-th layer, $T$ represents the total length of the sequence after concatenating speech and text, and $M$ denotes the total number of speech and text experts combined, respectively. $f(x) = W \cdot x$ is the linear modality router to produce expert assignment probabilities and $W \in R^{h \times d}$. $h$ is the last dimension of hidden states and $d$ is the total number of experts.

**Audio-Text Modeling** Following parallel modeling paradigm Défossez et al. (2024); Xie & Wu (2024); Chen et al. (2024); Gao et al. (2025), we use different heads to process hidden states, generating both text and audio tokens Chen et al. (2024); Défossez et al. (2024) in one decoding step. Since the input to the LLM is the average of the text and audio representations instead of native text representations, maintaining the original capabilities of the LLM can be challenging. To enhance models' text ability, we apply batch parallel decoding Xie & Wu (2024); Gao et al. (2025), Batch-parallel decoding expands a single input to a batch size of two: the first sample generates text-only response and the second sample generates both text and speech response. The generated text tokens from the first sample are of better quality than those from the second and are used to substitute the latter in the second sample.

**Audio Codec Decoder** We use SNAC codec Siuzdak et al. (2024) to encode the output speech into discrete tokens using 7 codebooks, which have a total token rate of 82 Hz. Combining speech and text tokens results in eight layers of labels, so we use eight LM heads to simultaneously predict one text token and seven audio token at each decoding step. We follow the delay pattern of MusicGen (Copet et al., 2024) for better generation quality, applying $k$ token delay to $k$-th layer of audio tokens.

---

**Algorithm 1** Adaptive Modality Expert Partitioning

---

**Require:** • Multimodal dataset with audio input: $\mathcal{D}^A = \{(x_i^A, y_i^{A,T})\}_{i=1}^n$
    • Multimodal dataset with text input: $\mathcal{D}^T = \{(x_j^T, y_j^{A,T})\}_{j=1}^m$
    • Pre-trained MoE-LLM $\pi_\theta$ after modality alignment
**Ensure:** Specialized expert allocation strategy $\{\mathbf{E}_l^A, \mathbf{E}_l^T\}_{l=1}^L$ {$L$ is the total number of layers.}
 1: Initialize: $k \leftarrow$ # audio experts, $M \leftarrow$ # total experts, $C \leftarrow$ # activated experts
 2: **Phase 1: Expert Selection Statistics**
 3: Extract hidden states: $\mathbf{h}_l^A = \pi_\theta(\mathcal{D}^A)$, $\mathbf{h}_l^T = \pi_\theta(\mathcal{D}^T)$ {Layer-wise representations}
 4: Compute token counts: $T^A = |\mathcal{D}^A|$, $T^T = |\mathcal{D}^T|$
 5: Initialize counting matrices: $\mathbf{E}_l^A \leftarrow \mathbf{0}_M$, $\mathbf{E}_l^T \leftarrow \mathbf{0}_M$ {Per-layer expert counters}
 6: **for** $(modality, \mathbf{h}_l)$ in $\{(A, \mathbf{h}_l^A), (T, \mathbf{h}_l^T)\}$ **do**
 7:    **for** $i \leftarrow 1$ to $T^{modality}$ **do**
 8:       $\mathcal{E}_{l,i}^{modality} \leftarrow$ top-k$(\text{MoE}(\mathbf{h}_l^{modality})_i, C)$ {MoE using equation 3}
 9:       **for** $j \leftarrow 1$ to $\text{len}(\mathcal{E}_{l,i}^{modality})$ **do**
10:          $\mathbf{E}_l^{modality}[\mathcal{E}_{l,i}^{modality}[j]] \mathrel{+}= 1$ {Aggregate selections}
11:       **end for**
12:    **end for**
13: **end for**
14: **Phase 2: Modality Token Load Ratio Computation**
15: **for** $modality \in \{A, T\}$ **do**
16:    **for** $j \leftarrow 1$ to $M$ **do**
17:       $\rho_{l,j}^{modality} \leftarrow \frac{\mathbf{E}_{l,j}^{modality}}{C \cdot T^{modality}}$ {Token load ratio $\rho \in [0, 1]$}
18:    **end for**
19: **end for**
20: **Phase 3: Partitioning Modality Experts Based on Modality Token Load**
21: **for** $j \leftarrow 1$ to $M$ **do**
22:    Audio_Experts$_l \leftarrow$ top-k$(\rho_{l,j}^A * (1 - \rho_{l,j}^T), k)$
23:    Text_Experts$_l \leftarrow$ top-k$(\rho_{l,j}^T * (1 - \rho_{l,j}^A), M - k)$
24: **end for**
25: **return** $\pi_\theta(\text{Audio\_Experts}_l, \text{Text\_Experts}_l)$

---

## 3.2 Adaptive Modality-Specific Mixture-of-Experts

The adaptive modality expert selection strategy is primarily designed for an original textual LLM, dynamically distinguishing between audio experts and text experts based on modality load. The goal is to ensure minimal impairment to the language capabilities of the original LLM. Specifically, as shown in Algorithm 1, an initial MoE model that has undergone stage 1 modality alignment training is used. For unimodal input data, the token load rates of all experts for both modalities are calculated. According to Algorithm 1, those experts with low text token load but high audio token load are selected as audio experts, while the others are designated as text experts. This adaptive selection approach fully utilizes the text experts with low load in the original MoE LLM, while also ensuring that the selected experts are suitable for processing the audio modality. In this way, the strategy minimizes the impact on the original LLM while fitting the audio modality as much as possible.

## 3.3 Training Strategy for Modality Experts

In order to fully leverage the capabilities of both text and audio experts, it is necessary to conduct specialization training for the designated modality experts, enabling each group of modality experts to excel at processing inputs from their respective modalities. Finally, to handle multimodal information inputs simultaneously, cross-modal instruction data is used for joint training of the modality experts. The specific training process is illustrated in Figure 3.

**Modality Aligment** During the modality alignment stage, audio ASR data is used to align the semantic spaces of speech and text. In this stage, the main focus is on training the Adapter that connects the Audio Encoder and the LLM. Through modality alignment training, it is ensured that speech can be understood by the LLM in the same way as the meaning expressed by text. The specific approach is to use a cross-entropy loss to align the audio and text modalities. Because the audio sequence for the same content is significantly longer than the text, in addition to downsampling the audio, we also use text padding tokens to align their lengths.

**Unimodal Expert Specialization Training**   The main purpose of unimodal specialization training is to enable modality experts to focus more on their own modality domains, so that they can fully utilize their expertise when collaborating with other multimodal experts. Since audio, like text, covers a wide range of domains, there are also multiple audio experts, each responsible for learning specialized knowledge in different areas, such as pitch, prosody, dialect, speech rate, and emotion. As shown in step 2 of Algorithm 1, audio and text experts are specialized and trained on their respective modalities. Since only unimodal specialization learning is required, the router is kept frozen to prevent it from becoming biased toward a single modality. During the specialization of audio experts, the router weight scores corresponding to text experts are set to zero. The remaining router weight scores are then normalized and redistributed among the audio experts. The same process is applied when specializing the text experts.

**Joint Training of Modality Experts**   Multimodal joint training is mainly designed to enable the model to jointly train unimodal experts so that they can collaboratively process multimodal input information. As shown in Stage 3 of Figure 3, this stage uses cross-modal outputs from both modalities to jointly train the audio and text experts. Since all experts participate in training at this stage, the router is also unfrozen and trained, allowing it to better learn how to handle modality routing when processing multimodal joint inputs.

### 3.4 AUDIO GENERATION WITH REINFORCEMENT LEARNING

To broaden the coverage of speakers and domains, it is inevitable that the pre-training data contains label noise and pronunciation errors, which can lead to hallucinations in the model. We observed that the quality of generated response audio is not consistent under different sampling parameters. To ensure more stable quality in response audio generation, we introduce a reinforcement learning stage to improve the stability of speech generation. Specifically, we use the text from the LibriSpeech Panayotov et al. (2015) set and have the model paraphrase it to obtain response audio and corresponding text. The response audio is then transcribed into recognized text using Whisper-large, and the word error rate (WER) is calculated as the reward score. All samples are ranked according to their WER, and triplet preference data $(x, y_w^A, y_l^A) \sim D$ is constructed for direct preference optimization (DPO) Rafailov et al. (2023) training as follows:

$$L_{DPO}(P_\theta; P_{ref}) = -E_{(x, y_w^A, y_l^A) \sim D}[log\sigma(\beta log \frac{P_\theta(y_w^A|x)}{P_{ref}(y_w^A|x)} - \beta log \frac{P_\theta(y_l^A|x)}{P_{ref}(y_l^A|x)}], \quad (4)$$

where $\theta$ is the policy model and $ref$ is the reference model. $x$ denotes the text input of the audio to be synthesized, $y_w^A$ denotes a good speech sequence, and $y_l^A$ denotes a bad speech sequence.

## 4 EXPERIMENTAL RESULTS AND ANALYSIS

### 4.1 EXPERIMENT SETTINGS

**Models**   The overall architecture of the DeepOmni model is shown in Figure 2. We use the pre-trained DeepSeek-V2-Lite as the initial MoE-based LLM backbone, Whisper-medium as the audio

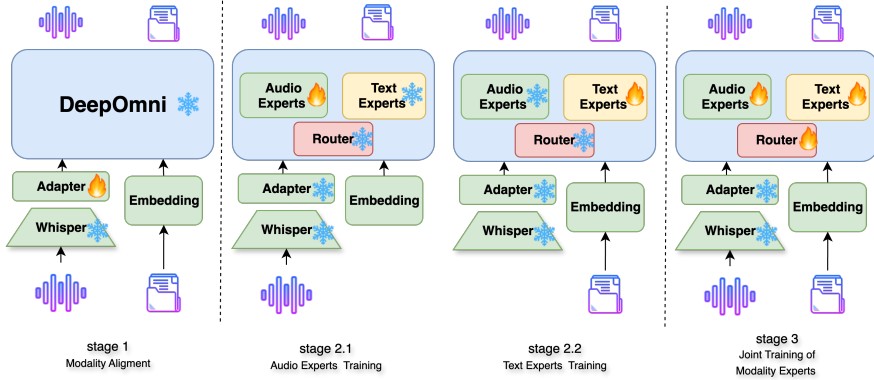

Figure 3: The training process consists of three stages: stage 1 is for modality alignment, stage 2 is for training single modality experts, and stage 3 is for joint training of the modality experts.

Table 2: Performance on Spoken Question Answering

| Model | #A-Param | LLaMA Question | | Web Question | | TriviaQA | | Avg | |
|---|---|---|---|---|---|---|---|---|---|
| | | S → T | S → S | S → T | S → S | S → T | S → S | S → T | S → S |
| Modular MLLMs | | | | | | | | | |
| Minmo Chen et al. (2025) | 7B | 78.9 | 64.1 | 55.0 | 39.9 | 48.3 | 37.5 | 60.7 | 47.2 |
| Step-Audio Huang et al. (2025) | 130B | 81.0 | - | 75.1 | - | 58.0 | - | 71.4 | - |
| Freeze-Omni Wang et al. (2024) | 7B | 72.0 | - | 44.7 | - | 53.9 | - | 56.9 | - |
| Native MLLMs | | | | | | | | | |
| Moshi Défossez et al. (2024) | 7B | 62.3 | 21.0 | 26.6 | 9.2 | 22.8 | 7.3 | 37.2 | 12.5 |
| GLM-4-Voice Zeng et al. (2024) | 9B | 64.7 | 50.7 | 32.2 | 15.9 | 39.1 | 26.5 | 45.3 | 31.0 |
| LUCY Gao et al. (2025) | 7B | 59.6 | 51.0 | 26.6 | 18.2 | 23.2 | 18.2 | 36.5 | 29.1 |
| MoExtend Zhong et al. (2024) | 2.4B | 54.0 | 45.3 | 20.8 | 16.9 | 18.5 | 17.2 | 31.1 | 26.5 |
| DeepOmni | 2.4B | 65.0 | 59.7 | 34.4 | 23.1 | 41.2 | 27.5 | 46.9 | 36.8 |

encoder, and an MLP-based adapter to connect the audio encoder and the LLM. DeepOmni contains a total of 66 experts, including 6 audio experts, 58 text experts, and 2 shared experts. A sparse routing mechanism is adopted, where each token is processed by 6 experts. The model consists of 27 layers in total: the first layer is a dense layer, followed by 26 MoE layers. Each MoE layer has a hidden size of 1408, and the self-attention layers use flash-attention Dao (2023) with 16 attention heads. In stage 1, the learning rate is set to 2e-5. Since the original LLM has not undergone any pre-training on audio modalities, the learning rate is set to 1e-4 in stage 2.1 to accelerate convergence, 2e-5 in stage 2.2, and 5e-5 in stage 3. We use the AdamW Loshchilov & Hutter (2017) optimizer with $\beta_1 = 0.9$ and $\beta_2 = 0.95$. The final evaluation is conducted on LLM benchmarks, SQA, ASR, and TTS tasks.

**Datasets** In stage 1, WenetSpeech Zhang et al. (2022) is used for modality alignment. In stage 2.1, AudioQA-1M Gao et al. (2025) is used for specialized training of audio experts. In stage 2.2, databricks-dolly-15k Dolly (2023), MathInstruct Yue et al. (2023) (262K) and camel-ai-math Li et al. (2023) (50K) are used for specialized training of text experts. In stage 3, AudioQA-1M is used for joint training of modality experts. For the RL stage, we sampled 28K text entries from LibriSpeech Panayotov et al. (2015) to construct the audio preference pair dataset.

## 4.2 EVALUATIONS

**Evaluation on Text to Text** The text evaluation is mainly conducted to assess the retention of language capabilities in the models, and all evaluations are performed using opencompass[1]. As shown in Table 3, modular MLLMs cause minimal damage to the language abilities of the original text LLMs, primarily because the LLM is only responsible for generating text tokens. However, native MLLMs significantly impair the language capabilities of the original text LLMs, with an average relative loss of over 20% compared to their backbone LLMs. The DeepOmni model effectively alleviates the catastrophic forgetting problem of native MLLMs through modality isolation and joint modality training, achieving an average relative loss at the same level as modular MLLMs.

**Evaluation on Speech to Text** The evaluation of speech-to-text is divided into two tasks: Spoken QA and ASR. As shown in the S → T column of Table 2, thanks to the preservation of the original LLM's language capabilities by modular MLLMs, these models generally achieve leading performance. In contrast, native MLLMs cause significant degradation to the original LLM, resulting in increased hallucinations and thus lower scores. However, DeepOmni effectively mitigates the damage caused by native multimodality to the LLM, preserving the language capabilities of the LLM to the greatest extent. As a result, DeepOmni leads among native MLLMs. The ASR capabilities of the models are shown in Table 4, where DeepOmni, with a relatively small number of parameters, achieves performance comparable to other models with much larger parameter sizes.

**Evaluation on Text to Speech** As shown in Table 1, the last three rows present the results for DeepOmni, where "topk" denotes the sampling parameter for audio tokens. Since the speech dialogue data used by DeepOmni contains a relatively high proportion of Chinese, the model performs better on Chinese tasks but is somewhat lacking in English. After optimizing audio generation during response with DPO, the model may already perform well on simple tasks, as DPO increases the probability of the correct audio appearing in pass@1, thereby making high-quality audio generation more stable. However, for more difficult tasks (test-hard), the supervised fine-tuning (SFT) model

---

[1] https://opencompass.org.cn/

Table 3: Performance of Text to Text on LLM benchmark, ∗ indicates that the average metric is calculated only on those test sets for which Qwen2.5-Omni has values. HS, WG, HEval denotes Hellaswag, Winogrande, HumanEval, respectively. "#AP" denotes number of activated parameters.

| Model | #AP | General | | | Reasoning | | | | | Math | | Coding | | Avg | Drop↓ |
|---|---|---|---|---|---|---|---|---|---|---|---|---|---|---|---|
| | | CEval | CMMLU | MMLU | BBH | ARC_c | GPQA | HS | WG | MATH | GSM8K | HEval | MBPP | | |
| | | | | | | | Modular MLLMs | | | | | | | | |
| Qwen2-7B | 7B | 81.62 | 80.79 | 70.76 | 64.72 | 85.42 | 34.30 | 79.15 | 66.93 | 52.90 | 82.87 | 79.17 | 67.60 | 70.52 | N/A |
| VITA-1.5 | 7B | 77.11 | 76.85 | 73.43 | 66.01 | 70.17 | 23.23 | 79.45 | 65.90 | 36.90 | 71.95 | 75.00 | 58.20 | 64.52 | -8.51 |
| Qwen2.5-7B | 7B | 78.74 | 78.85 | 74.19 | 70.06 | 88.14 | 36.40 | 82.40 | 68.43 | 75.50 | 91.60 | 84.80 | 79.20 | 73.62* | N/A |
| Qwen2.5-Omni | 7B | - | - | 71.00 | - | - | 30.08 | - | - | 71.50 | 88.70 | 78.70 | 73.20 | 68.86 | -6.47 |
| | | | | | | | Native MLLMs | | | | | | | | |
| GLM-4 | 9B | 74.48 | 72.78 | 69.93 | 69.82 | 92.20 | 21.21 | 84.60 | 76.80 | 37.52 | 75.89 | 75.00 | 61.20 | 64.08 | N/A |
| GLM-4-Voice | 9B | 39.64 | 54.76 | 52.48 | 29.02 | 63.05 | 25.76 | 40.85 | 47.20 | 3.80 | 32.37 | 26.61 | 12.60 | 35.68 | -44.32 |
| Qwen2-0.5B | 0.5B | 54.71 | 47.59 | 42.99 | 27.94 | 44.41 | 1.01 | 31.70 | 49.88 | 11.48 | 35.94 | 28.66 | 20.20 | 33.04 | N/A |
| Mini-Omni | 0.5B | 32.53 | 28.64 | 34.25 | 20.13 | 32.18 | 0.67 | 25.93 | 36.62 | 8.67 | 30.48 | 21.79 | 18.63 | 24.21 | -26.73 |
| Qwen2-7B | 7B | 81.62 | 80.79 | 70.76 | 64.72 | 85.42 | 34.30 | 79.15 | 66.93 | 52.90 | 82.87 | 79.17 | 67.60 | 70.52 | N/A |
| LUCY | 7B | 53.38 | 52.76 | 59.43 | 47.48 | 78.31 | 14.14 | 76.30 | 58.88 | 24.24 | 68.92 | 46.34 | 36.40 | 51.38 | -27.14 |
| DeepseekV2-Lite | 2.4B | 58.60 | 62.06 | 57.56 | 49.20 | 75.25 | 23.74 | 67.90 | 61.40 | 18.98 | 59.67 | 57.32 | 45.00 | 53.06 | N/A |
| DeepOmni | 2.4B | 58.45 | 59.05 | 54.63 | 48.27 | 68.98 | 20.71 | 63.25 | 59.59 | 17.68 | 55.40 | 54.67 | 41.33 | 50.17 | -5.45 |

Table 4: Performance of Speech to Text on ASR. The metric for Chinese is CER, and for English WER. Lower values are better.

| Model | #A-Params | WenetSpeech | | AIShell | LibriSpeech | |
|---|---|---|---|---|---|---|
| | | test-net | test-meeting | test | test-clean | test-other |
| Qwen2-Audio-base Chu et al. (2024) | 7B | 7.64 | 8.40 | 1.52 | 1.74 | 4.04 |
| Baichuan-Audio-base Li et al. (2025a) | 7B | 10.13 | 13.28 | 1.93 | 3.02 | 6.04 |
| VITA Fu et al. (2025) | 7B | 12.15 | 16.53 | - | 8.14 | 18.41 |
| Step-Audio-chat Huang et al. (2025) | 130B | 9.47 | 10.83 | 2.14 | 3.19 | 10.67 |
| Qwen2.5-Omni Xu et al. (2025) | 7B | 6.04 | 7.71 | 1.13 | 2.37 | 4.21 |
| GLM-4-Voice Zeng et al. (2024) | 9B | - | - | 3.02 | 2.10 | 4.90 |
| DeepOmni | 2.4B | 8.98 | 10.67 | 2.42 | 3.25 | 8.21 |

itself may not generate the correct result within the entire search space, so applying DPO brings little to no benefit.

**Evaluation on Speech to Speech** The evaluation of speech-to-speech is shown in the S → S column of Table 2. MoExtend refers to the results obtained by extending the original Deepseek-V2-Lite model with additional experts, adding six audio experts. However, since expanding the number of experts alters the distribution of the router, the performance is relatively suboptimal. DeepOmni, on the other hand, adopts an adaptive modality expert selection strategy, preserving the original router's allocation to experts. This allows the audio experts to specialize while retaining the capabilities of the original text experts, thus achieving better performance among native MLLMs.

**Latency** We deployed DeepOmni on a web server and measured end-to-end latency in half-duplex mode, where users manually control the timing of their speech input, without involving a Voice Activity Detection (VAD) module. The reported numbers are averaged over more than 10 samples. With a good network connection, the end-to-end latency of the entire system in half-duplex mode is 0.4362 seconds. The time cost for generating the first audio chunk is 0.3417 seconds. By subtracting the first chunk cost from the total latency, we estimate the network transmission cost to be 0.1062 seconds. The time cost for decoding a single token is 0.0205 seconds. All time measurements were conducted on a GPU with 80GB of memory.

## 5 CONCLUSION

To address the catastrophic forgetting problem caused by the scarcity of paired data, we propose DeepOmni, a MoE framework for adaptive modality expert learning. DeepOmni first adaptively distinguishes modality experts according to their modality load within the LLM. Each modality expert then undergoes specialized single-modality training, followed by joint multimodal collaborative training. As a result, DeepOmni incurs a performance loss lower than peer native MLLMs and on par with modular ones. Meanwhile, the end-to-end dialogue latency remains within 0.5 seconds, ensuring a seamless and intelligent speech interaction experience. To the best of our knowledge, this is the first MoE-based native multimodal large speech interaction model, which opens up a new direction for mitigating catastrophic forgetting in native multimodal speech interaction models and bridges the gap between native multimodal and modular multimodal approaches.

## 6 REPRODUCIBILITY STATEMENT

We have released all reproducible code in the supplementary materials and anonymized it. The details of the training data used and the hyperparameters are provided in Appendices A.1 and B, respectively.

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

# A APPENDIX

## A.1 DATASET DETAILS

As shown in Table 5, we provide the data and quantities used for training at each stage. Stage 1's training data consists of speech-text pairs, which are used to align the semantic spaces of the speech and text modalities. AudioQA-1M contains conversational instruction data, with both input and output encompassing two modalities, aimed at fine-tuning the model to enhance its speech dialogue capabilities. Stage 2.2 utilizes pure text data to maintain the language proficiency of the text expert. During the RL stage, data from Librispeech is used to construct DPO preference pair data for speech generation.

Table 5: List of datasets used for each training stage.

| Stage | Dataset | Size |
|---|---|---|
| 1 | WenetSpeech | 10Kh |
| 2.1 | AudioQA-1M | 1000K |
| 2.2 | MathInstruct | 262K |
| | camel_ai_math | 50K |
| | databricks_dolly_15k | 15K |
| 3 | AudioQA-1M | 1000K |
| | MathInstruct | 262K |
| | camel_ai_math | 50K |
| | databricks_dolly_15k | 15K |
| RL | LibriSpeech-DPO | 28K |

# B EXPERIMENTAL SETUP DETAILS

Table 6 outlines the main hyperparameters used during each stage of training. All ablation studies were conducted using a total batch size of 128, with learning rates of 1e-4, 2e-5, and 5e-5.

Table 6: Hyperparameters used in each stage training on DeepOmni.

| Hyperparameter | Stage 1 | Stage 2.1 | Stage 2.2 | Stage 3 | Stage RL |
|---|---|---|---|---|---|
| Learning rate | 2e-5 | 1e-4 | 2e-5 | 5e-5 | 2e-5 |
| LR schedule | Cosine | Cosine | Cosine | Cosine | Cosine |
| Batchsize per GPU | 16 | 8 | 8 | 8 | 8 |
| Zero | Zero2 | Zero2-offload | Zero2 | Zero2-offload | Zero3 |
| Optimizer | AdamW | AdamW | AdamW | AdamW | AdamW |

## B.1 ABLATION STUDY

### B.1.1 AUDIO EXPERTS NUMBER

We conducted ablation experiments with varying numbers of audio experts, as shown in Table 7. The results demonstrate that as the number of audio experts increases, the language ability tends to be compromised, whereas the speech dialogue capability is enhanced. However, once the number of audio experts surpasses 24, both speech dialogue and language abilities begin to decline. This suggests that at this point, too many resources are allocated to audio experts at the expense of the text experts of the original text LLM, leading to a decrease in language ability. Moreover, the speech dialogue capability also falls after reaching a bottleneck concerning language ability.

Table 7: Ablation study on the impact of the number of audio experts on the capabilities of the two modalities.

| Experts Num | LLaMA Question | | Web Question | | TriviaQA | | Avg | |
|---|---|---|---|---|---|---|---|---|
| | T → T | S → S | T → T | S → S | T → T | S → S | T → T | S → S |
| 6 | **69.0** | 59.7 | **46.6** | 23.1 | **58.2** | 27.5 | **57.9** | 36.8 |
| 12 | 61.0 | **60.0** | 39.2 | 26.7 | 49.7 | 29.1 | 50.0 | 38.6 |
| 18 | 59.0 | 59.0 | 32.1 | **28.3** | 42.6 | **33.7** | 44.6 | **40.3** |
| 24 | 48.0 | 48.0 | 23.7 | 22.9 | 31.9 | 30.2 | 34.5 | 33.7 |

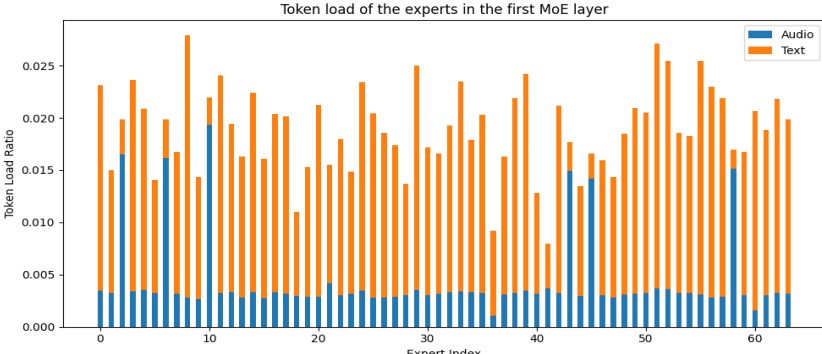

Figure 4: The modality load of the experts in the first layers.

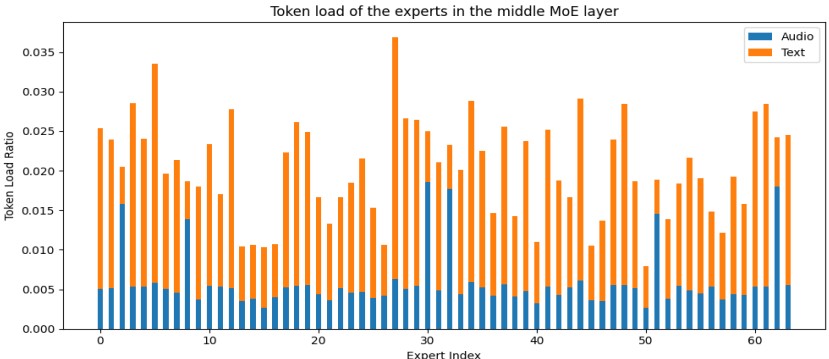

Figure 5: The modality load of the experts in the middle layers.

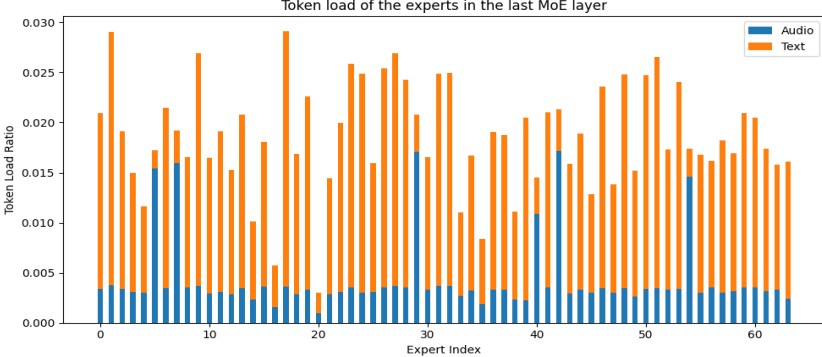

Figure 6: The modality load of the experts in the last layers.

Table 8: Performance on Spoken Question Answering

| Model | LLaMA Question | | Web Question | | TriviaQA | | Avg | |
|---|---|---|---|---|---|---|---|---|
| | S $\rightarrow$ T | S $\rightarrow$ S | S $\rightarrow$ T | S $\rightarrow$ S | S $\rightarrow$ T | S $\rightarrow$ S | S $\rightarrow$ T | S $\rightarrow$ S |
| PureMoE | 48.3 | 40.7 | 20.9 | 15.7 | 21.5 | 17.6 | 30.2 | 24.7 |
| LoRA-PureMoE | 50.4 | 37.6 | 22.3 | 13.8 | 23.1 | 14.1 | 31.9 | 21.8 |
| Random Modality-Specific(1) | 40.3 | 32.7 | 17.3 | 10.9 | 18.9 | 14.5 | 25.5 | 19.4 |
| Random Modality-Specific(2) | 54.3 | 45.7 | 25.9 | 19.8 | 26.3 | 20.1 | 35.5 | 28.5 |
| Random Modality-Specific(3) | 52.1 | 47.0 | 23.3 | 17.6 | 29.8 | 23.4 | 35.1 | 29.3 |
| Adaptive Modality-Specific | **65.0** | **59.7** | **34.4** | **23.1** | **41.2** | **27.5** | **46.9** | **36.8** |

### B.1.2 THE MODALITY LOAD OF DIFFERENT LAYER EXPERTS

The expert loads at different layers are shown in Figures 4, 5, and 6, corresponding to the modality loads of the first, middle, and last MoE layers, respectively. We can see that the modality loads differ substantially across layers, indicating that the indices of the audio experts vary from layer to layer. Moreover, the modality token loads at each layer also reveal the specialization of the modality-specific experts.

### B.1.3 MODALITY-SPECIFIC MOE

As shown in Table 8, we compared two settings: PureMoE without specifying modality experts and Random Modality-Specific MoE with randomly assigned experts. We found that if modality experts are not set, the model can only rely on self-learning the modality boundaries. During the early stages of training, the model does not clearly understand these boundaries, and the inclusion of audio data may significantly harm important text experts, leading to poorer final performance. The random assignment of modality experts was attempted three times, but results were inconsistent. If suitable experts were chosen as audio experts, the performance could be quite good, as seen in the second and third instances. However, if some crucial text experts were chosen as audio experts, the initial LLM suffers substantial damage, resulting in noticeable performance degradation. LoRA-PureMoE builds on PureMoE by applying LoRA-based fine-tuning to the LLM component. Although this mitigates the degradation of the LLM's text capabilities, the speech modality learning remains insufficient. Using an adaptive modality assignment method effectively selects which experts are suitable as audio experts and text experts, achieving better performance.

### B.2 BROADER IMPACT

DeepOmni leverages pre-trained LLMs, which are inherently subject to the limitations of LLMs. These limitations include the potential to generate inaccurate information or biased outputs. To address these issues, we enhanced the model's speech interaction capabilities using AudioQA and performed speech-language instruction tuning on high-quality datasets. Despite these improvements, we recommend exercising caution and conducting thorough safety and fairness evaluations before deploying the DeepOmni model in any downstream applications.

## C THE USE OF LARGE LANGUAGE MODELS

We used LLMs for grammar correction and proofreading.

