# OpenReview forum: "DeepOmni: Towards Seamless and Smart Speech Interaction with Adaptive Modality-Specific MoE"
_ICLR.cc/2026/Conference — Submitted to ICLR 2026_

### Official Review · Reviewer_YCDH · 2025-10-31

**Soundness:** 3
**Presentation:** 3
**Contribution:** 2
**Rating:** 4
**Confidence:** 4

**Summary:**

The paper introduces DeepOmni, an MoE-based speech interaction model built on the DeepseekV2-Lite backbone. It follows a parallel modeling paradigm and employs the SNAC codec for speech tokenization. To address the catastrophic forgetting issue in native MLLMs, DeepOmni adaptively identifies modality-specific experts based on modality load, performs specialized single-modality training, and concludes with joint multimodal collaborative training. Experiments on text-to-text tasks demonstrate only a 5.5% performance drop compared to the original LLM.

**Strengths:**

1. The paper proposes the first MoE-based native speech interaction model, effectively addressing the catastrophic forgetting issue in native MLLMs, which is a critical research topic.
2. The adaptive modality-specific MoE design is innovative and is supported by ablation studies showing its advantages over MoExtend, PureMoE, LoRA, and Random Modality-Specific approaches.
3. The paper is written clearly and includes code in the supplementary material, enhancing reproducibility.

**Weaknesses:**

1. The paper uses batch parallel decoding, which increases computational costs and creates an unfair comparison with baselines that do not use this technique. Results without batch parallel decoding should be provided.
2. The research is based on a relatively weak backbone, making it unclear how the model would perform with stronger backbones like Qwen3-30B-A3B.
3. There is a lack of comparison with key baselines in Table 2&3 such as Kimi-Audio and Step-Audio2-Mini, which are also native MLLMs. Table 2 suggest that DeepOmni underperforms these models on Spoken QA.

Minor:
1. GPU model details should be added for latency testing.

**Questions:**

Please address the issues described in the Weaknesses section. Resolving these concerns could improve the paper’s evaluation.

---

> ### Author Response · Authors · 2025-11-20
>
> **Q1:** This paper employs batch parallel decoding, which increases computational costs, leading to an unfair comparison when benchmarked against baseline methods that do not use this technique. Results without batch parallel decoding should be provided.
>
> **A1:** Batch parallel decoding does not increase computational costs; on the contrary, it accelerates inference and reduces costs. Without batch parallel decoding, speech generation latency would be very high. For the same generated speech tokens, more forward passes would be required, resulting in greater computational cost. Currently, mainstream end-to-end native MLLM speech-text modeling approaches are broadly divided into two categories: parallel and interleaved. The advantage of parallel decoding is that multiple tokens can be decoded in each forward pass. Since audio tokens significantly outnumber text tokens for the same content, generating multiple audio tokens per step is very beneficial. The advantage of interleaved decoding is that it can, to some extent, preserve the "intelligence" of the LLM.
>
> ---
>
> **Q2:** This research is based on a relatively weaker backbone network, so it remains unclear how the model would perform on a more powerful backbone like Qwen3-30B-A3B.
>
> **A2:** Thank you very much for your suggestion. At the time, there wasn't a suitable MoE foundation model (small in parameters yet strong in performance), so we chose DeepSeek-V2-Lite, which was more appropriate then. However, based on comparisons with PureMoE, our method's effectiveness should be quite significant. We also look forward to validating the results on newer, more powerful foundation models.
>
> ---
>
> **Q3:** Tables 2 and 3 lack comparisons with key baseline models such as Kimi-Audio and Step-Audio2-Mini, which are also native Multimodal Large Language Models (MLLMs). Table 2 shows that DeepOmni underperforms these models on speech question-answering tasks.
>
> **A3:** Although Kimi-Audio appears to be native end-to-end, it actually is not. Its audio head consists of transformer layers, not the embedding layer (LLM head) of the LLM. In practice, the audio head is handled by a separate module (similar to a speech LLM decoder) responsible for generating audio tokens, making it essentially a modular MLLM.
> Step-Audio2-Mini is a native MLLM, but it was released slightly later. We tested Step-Audio-Mini on S -> T, and its performance is indeed better. We speculate this is primarily due to the weaker LLM backbone of DeepOmni. We may report the metrics after conducting experiments on Qwen3-30B-A3B.
>
> | Model | LLaMA Question | Web Question | TriviaQA |
> | :--- | :--- | :--- | :--- |
> | DeepOmni(DeepSeek-V2-Lite-A2.4B) | 65.0 | 34.4 | 41.2 |
> | Step-Audio2-Mini-7B | 69.0 | 42.4 | 50.3 |
>
> ---
>
> **Q4:** GPU model details should be added for latency testing.
>
> **A4:** We sincerely apologize, but due to company policy, we are not permitted to disclose the specific GPU models used.

---

> > ### Comment · Reviewer_YCDH · 2025-11-24
> >
> > Some questions remain:
> >
> > Q1: I’m referring to the batch-parallel decoding mentioned in lines 261–265. Could you please elaborate on how this is implemented in DeepOmni?
> >
> > Q2: I look forward to seeing your metrics after running experiments on Qwen3-30B-A3B.

---

> > > ### Author Response · Authors · 2025-11-25
> > >
> > > Thank you for your reply. Regarding the first question, splitting a batch into two samples does indeed increase computational overhead, but this overhead is mainly reflected in latency. Since it's batch decoding, it essentially just doubles the batch size (1→2), and the latency increase is almost negligible. Additionally, the main idea behind this approach is to prevent the generated text tokens from being interfered with by other modalities, resulting in more stable text generation quality. On knowledge QA benchmarks like LLaMA Question/Web Question/TriviaQA, adding batch parallel decoding does not show any metric changes. However, it might bring some performance improvements in other scenarios that are more reasoning-oriented.
> > >
> > > The main implementation flow of batch parallel decoding can be explained with the following pseudocode:
> > > ```
> > > for t in range(max_length):
> > >     next_t, next_a = next_token(model, input_ids)
> > >     next_t[1] = next_t[0]  # For the second sample generating text and audio, the text token is replaced by the first sample's text token.
> > >     next_a[0, :] = PAD_A   # For the first sample generating text only, the audio token part is padded with PAD.
> > >     input_ids = torch.cat([next_a, next_t])
> > >     if next_t[0, -1] == EOT and next_a[1, -1] == EOA:
> > >         break
> > > ```

---

> > > > ### Author Response · Authors · 2025-11-26
> > > > **Experiments Update**
> > > >
> > > > For the second question, the metrics for Qwen3-30B-A3B have been updated. It can be observed that with the upgrade to a stronger base model, the model's performance has improved accordingly. Additionally, a comparison was made between Step-Audio2-Mini and Kimi-Audio. To maintain the foundational general capabilities of the text model, Step-Audio2-Mini incorporated 400B text tokens during the pre-training and SFT stages. Kimi-Audio, being not a purely native MLLM but rather a Modular MLLM, exhibits stronger foundational general capabilities. However, as a native MLLM, the DeepOmni model still maintains decent general text capabilities despite being trained on a limited number of text tokens.
> > > > | Model | LLaMA Question | Web Question | TriviaQA | E2E Type |  Training Stage | Text Tokens | Text Conversations Num |
> > > > | :--- | :--- | :--- | :--- |:--- | :--- | :--- |:--- |
> > > > | DeepOmni(DeepSeek-V2-Lite-A2.4B) | 65.0 | 34.4 | 41.2 | Native |  SFT | 264M | 1.5M |
> > > > | Step-Audio2-Mini-7B | 69.0 | 42.4 | 50.3 | Native |  Pretrain + SFT | 400B | - |
> > > > | Kimi-Audio-7B |  75.3 | 62.8 | 57.1 | Modular | Pretrain + SFT | - | 12.7M |
> > > > | DeepOmni(Qwen3-30B-A3B) | 73.0 | 48.3 | 52.1 | Native | SFT | 264M | 1.5M |

---

> > > > > ### Comment · Reviewer_YCDH · 2025-11-28
> > > > >
> > > > > Thank you to the authors for addressing my main concerns. I will raise my score from 4 to 6.

---

> > > > > > ### Author Response · Authors · 2025-11-28
> > > > > >
> > > > > > We are very pleased to have addressed your main concerns and achieved an improvement in the score. Thank you for your detailed review and time.

---

### Official Review · Reviewer_yAcB · 2025-11-01

**Soundness:** 2
**Presentation:** 1
**Contribution:** 2
**Rating:** 4
**Confidence:** 4

**Summary:**

This paper proposes DeepOmni, a native multimodal large language model for speech interaction that leverages Mixture of Experts (MoE) architecture to mitigate catastrophic forgetting. The key contribution is an adaptive modality expert selection strategy that dynamically assigns experts to audio or text modalities based on their "modality load." The model outperform other native MLLMs like GLM-4-Voice on text-to-text, speech-to-text and speech-to-speech tasks.

**Strengths:**

1. Addresses Important Problem: Catastrophic forgetting in native multimodal speech models is a genuine and pressing challenge. The paper tackles a real bottleneck that limits the practical deployment of end-to-end speech interaction systems.
2. Novel Adaptive Selection Strategy: The adaptive modality expert partitioning based on modality load is creative and well-motivated. Unlike random assignment, this data-driven approach intelligently identifies which experts are suitable for audio vs. text.
3. Comprehensive Evaluation: The experimental evaluation is thorough, covering multiple dimensions: spoken QA, ASR, TTS, and LLM benchmarks.

**Weaknesses:**

1. Weak Baselines in Comparison:
The results section appears to compare against relatively weak baselines. Why do Tables 2–5 not include comparisons with Qwen-2.5-OMNI and Kimi-Audio? Notably, Kimi-Audio is itself a non-modular speech LLM, making it an important baseline for fair evaluation.

2. Questionable Claims About Modular SLM Limitations:
The paper’s claims regarding the limitations of Modular Speech Language Models are not fully substantiated. These models remain end-to-end differentiable, for instance, Qwen-OMNI can leverage its generated LLM hidden representations to encode paralinguistic cues. Did the authors perform any experiments showing that modular architectures are indeed worse at modeling such paralinguistic information?


3. Lack of Analysis:
While the method demonstrates improved performance, there is limited insight into why it mitigates catastrophic forgetting more effectively than other methods. What distinct knowledge patterns do the audio and text experts capture? How does modality isolation help preserve capabilities? A deeper analysis—e.g., via probing or visualization—would greatly strengthen the paper.


4. Clarity and Presentation Issues: The paper is difficult to follow in several sections. The term e(h_l)_i should be explicitly defined in Eq. (2). Algorithm 1, which seems central to the contribution, is hard to interpret and should be explained in greater detail. The Expert Selection Statistics remain unclear. The multiplicative selection criterion (lines 22–23) seems arbitrary—why this specific formulation and not other combinations? An ablation or stronger motivation is needed. Formatting in Sections 3.3 and 3.4, as well as reference styling (perhaps using \citep{}), should also be improved.


5. Section 3.3 (Audio–Text Alignment):
The paper mentions “downsampling the audio and using text padding tokens to align their lengths.” More details should be provided—specifically, how much padding is applied and whether it affects training stability or convergence.

**Questions:**

Please check weaknesses, particularly 1,2 and 4

---

> ### Author Response · Authors · 2025-11-17
>
> We thank all the reviewers and meta-reviewers for the careful reviews and valuable comments. We will respond to each comment in the sections below according to each reviewer.
>
> ---
>
> **Q1:**
>
> Neither Qwen-2.5-OMNI nor Kimi-Audio are strictly native end-to-end MLLMs, so the comparison is not entirely fair.
>
> Although Kimi-Audio appears to be native end-to-end, it actually is not. Its audio head consists of transformer layers, not the embedding layer (LLM head) of the LLM. In practice, the audio head is handled by a separate module (similar to a speech LLM decoder) responsible for generating audio tokens, making it essentially a modular MLLM.
>
> **The experimental results of replacing the base model with a stronger version, Qwen3-30B-A3B, can be found in the response to Reviewer YCDH.**
>
> ---
>
> **Q2:**
>
> Modular MLLMs can indeed utilize hidden states to encode paralinguistic information. However, encoding each type of paralinguistic information requires a separate hidden state, which is overly cumbersome. We hope the model can natively support the output of paralinguistic information.
>
> ---
>
> **Q3:**
>
> Thank you for your suggestion.
>
> ---
>
> **Q4:**
>
> e(f(h_l))_i refers to the exponential e applied after the
> l-th layer's hidden state passes through the router, where
> i denotes the i-th token in the total input tokens.
>
> I apologize for any confusion regarding the expert selection algorithm.
>
> The core idea of the algorithm can be summarized as follows: For text modality experts, experts with high load in the text modality and low load in the audio modality should be selected as text modality experts. The selection of audio experts follows a similar principle. Overall, the formula is designed to fulfill the above requirements, and whether multiplication, addition, or another operation is used makes little difference.
>
> ---
>
> **Q5:**
>
> Fill the text tokens to the same length as the speech tokens. No impact on training stability or convergence has been observed so far.

---

### Official Review · Reviewer_mQes · 2025-11-01

**Soundness:** 3
**Presentation:** 3
**Contribution:** 3
**Rating:** 6
**Confidence:** 3

**Summary:**

The paper presents DeepOmni, a multimodal spoken language model that integrates adaptive modality-specific experts within a Mixture-of-Experts (MoE) architecture. The goal is to alleviate catastrophic forgetting in native multimodal large language models (MLLMs). DeepOmni introduces an adaptive modality expert selection strategy based on modality token load and employs a three-stage training procedure of modality alignment, unimodal training, and cross-modal joint training. Experiments on spoken QA, ASR, TTS, and text benchmarks demonstrate that DeepOmni reduces language performance degradation to 5.5%, substantially lower than existing native MLLMs (typically over 20%), while maintaining real-time response latency (<0.5 s). Overall, the paper contributes a novel and well-engineered MoE-based framework for building end-to-end speech interaction models that effectively balance linguistic competence and acoustic generation.

**Strengths:**

1. The work claims to be the first native MLLM built upon an MoE-based LLM backbone with a 3-stage post-training and addresses the catastrophic forgetting in native MLLM. Solid and highly effective.

2. It proposes an effective and intuitive expert partition strategy that selects modality-specific experts based on modality load, and the proposed model achieves a low performance drop in language capacity.

**Weaknesses:**

1. The paper claims native MLLMs preserve richer paralinguistic features as part of its motivation, but the evaluation lacks essential quality-based metrics to substantiate this claim and compare the expressive quality of the proposed model against other native baselines.

**Questions:**

1. See weakness 1. Can we see results comparing DeepOmni's speech output against other native MLLMs on quality metrics like prosody and emotional expression?

2. The process for designating the 2 shared modality experts is missing from the adaptive partitioning mechanism (Algorithm 1). Can the authors clarify this step?

3. The Phase 3 pseudo-code in Algorithm 1 puts $\text{top-}k$ inside a loop iterating over $j$. This looks confusing, as the intent seems to be applying $\text{top-}k$ globally, like \text{Audio Experts}_{l} \leftarrow \text{top-}k\!\left(
    \left\{ \rho_{l,j}^{A} \cdot (1 - \rho_{l,j}^{T}) \right\}_{j=1}^{M},\, k
\right)

4. Some formatting issues for inline citations. Some should be using citep

---

> ### Author Response · Authors · 2025-11-21
>
> We thank all the reviewers and meta-reviewers for the careful reviews and valuable comments. We will respond to each comment in the sections below according to each reviewer.
>
> **Q1:** Can we see results comparing DeepOmni's speech output against other native MLLMs on quality metrics like prosody and emotional expression?
>
> **A1:**
> Emotion-Aware Response Test Results (Internal Test Set):
>
> | Model  |  Neutral | Joy	| Anger | Fear | Disgust | Sadness | Surprise |
> | :--- | :--- | :--- | :--- |:--- |:--- |:--- |:--- |
> | GLM-4-Voice | 64.05 | 78.23	| 11.19 | 5.79 | 5.81 | 57.48 | 20.51 |
> | DeepOmni | 97.31	| 95.63 | 90.23	| 86.54 | 68.79	| 81.76 | 79.18 |
>
>
> ---
>
> **Q2:**  The process for designating the 2 shared modality experts is missing from the adaptive partitioning mechanism (Algorithm 1). Can the authors clarify this step?
>
> **A2:** All modality tokens are assigned to the 2 shared modality experts. Therefore, these 2 shared experts do not participate in the modality expert partitioning process and are consequently not listed in Algorithm 1. We will revise the algorithm by denoting M as the total number of experts excluding the shared ones.
>
> ---
>
> **Q3:** The pseudo-code for the third stage in Algorithm 1 places the top-k operation inside a loop iterating over j. This seems confusing.
>
> **A3:** The third stage iterates through all experts to filter and add those meeting the criteria to the Audio/Text expert lists. The outer for loop should indeed be removed so that the top-k selection is applied globally. Thank you for pointing this out.
>
> ---
>
> **A4:** Thank you for noting the citep format issue. We will correct it.

---

### Official Review · Reviewer_Vviv · 2025-11-01

**Soundness:** 3
**Presentation:** 2
**Contribution:** 2
**Rating:** 4
**Confidence:** 5

**Summary:**

To mitigate catastrophic forgetting in native MLLMs, this paper proposes DeepOmni for adaptive modality expert learning in a MoE-based MLLM. DeepOmni goes through stages of adaptive modality expert selection based on modality load, specialized single-modality training with instruction data from different modalities, and then joint multimodal collaborative training using cross-modal instruction data. Experimental results show that DeepOmni achieves a 5.5% relative performance drop over the original LLM, substantially lower than some MLLMs such as GLM-4-voice. The E2E dialogue latency remains 0.5 secs for efficient voice interaction.

**Strengths:**

1.	Using a MoE architecture for developing MLLMs has been explored in earlier works, as well as dynamic modality expert selection such as in prior works of LLMoE etc. The single-modality expert training and then cross-modal expert training has also been explored in the prior Uni-MoE framework. The main contribution of this work seems to investigate the impact of these previously proposed approaches on mitigating catastrophic forgetting of text capabilities in LALMs and omni models, which is an important research question. The experimental results show that DeepOmni achieves a 5.5% relative performance drop over the original LLM, which is better than the 6.5% relative drop from Qwen2.5-omni (a dense model) over its backbone LLM.

2. The analysis of the number of audio experts, modality load of experts at different layers are useful.  The comparison between PureMoE, LoRA-PureMoE,  Random Modality-specific MoE, Adaptive Modality-specific MoE shows clear advantages of adaptive modality-specific MoE over the less principled approaches.

**Weaknesses:**

1.	Some important related, non-contemporaneous works are missing in theoretical and empirical comparisons, for example, strong MLLMs such as Kimi-audio, Ming-lite-omni (which is also a MoE-based omni model). Hence, the presentation of the experimental results is misleading. For example, in Table 2 performance on Spoken QA,  Kimi-audio and Qwen2.5-omni achieved much better performance than the proposed DeepOmni, yet their evaluation results are missing. In Table 3 evaluating the T2T performance and the relative drop of the LALMs/omni models, Kimi-audio and other dense or MoE-based MLLMs are missing, such as Uni-MoE, Ming-lite-omni.

2. DeepOmni is built upon a weak backbone, DeepSeek-V2-Lite, which is further verified by its poor performance on text capabilities as shown in Table 3. As a 15.7B-A2.4B MoE model, its average score is 53.06, much worse than qwen2-7B's 70.52, qwen2.5-7B's 73.62, and GLM-4-9B's 64.08, with all these being dense models. With a low performing backbone, it is difficult to fully justify the effectiveness of the proposed  dynamic modality expert selection, uni-modality expert training and cross-modal expert training. It is important to evaluate these proposed approaches on a more competitive MoE backbone.

3. The batch parallel decoding used in DeepOmni, as also used in mini-omni and other works,  expands a single audio input into a batch size of two, with one audio+text sample, and one text-only sample, and embed the text-only output into the audio generation process. This is more a hybrid workaround rather than a principled solution for the speech and text interference in parallel speech-text modeling.

4. For S2S, the speech generation performance needs to be evaluated, for example, reporting WER and UTMOS.

5.	The ablation study in Appendix focuses on investigating the number of acoustical experts and analysis of modality load across different layers showing the benefit of dynamic modality expert selection, but the analysis of the multi-stage training is not presented.

**Questions:**

1.	There are some formatting issues. For example, the citations could be added using \citep, so that it would appear as, for example, (Radford et al., 2023). The current citation formatting right after text, e.g., (ASR) Radford et al. (2023), degrades readability.

---

> ### Author Response · Authors · 2025-11-16
>
> We thank all the reviewers and meta-reviewers for the careful reviews and valuable comments. We will respond to each comment in the sections below according to each reviewer.
>
> **Q1:**
> Kimi-Audio, Qwen2.5-omni, and Ming-lite-omni are all modular MLLMs, not true native end-to-end MLLMs. Uni-MoE can't even be called end-to-end; it only has text output and no speech output.
> Among them, Qwen2.5-omni adopts the Thinker-Talker architecture, where the Thinker is dedicated to generating text tokens and the Talker is responsible for generating audio tokens. This is significantly different from a true native end-to-end MLLM.
> Kimi-Audio appears to be a native end-to-end model but is actually not. Its audio head consists of transformer layers, not the embedding layer (LLM head) of the LLM. In fact, the audio head uses a separate module (similar to a speech LLM decoder) to handle audio token generation, making it essentially a modular MLLM.
>
> Since our DeepOmni model is a native end-to-end MLLM, comparing it with the above modular MLLMs is not entirely fair. Therefore, we have selected baselines that are all representative native end-to-end MLLMs.
> Additionally, due to the performance limitations of the base model DeepSeek-V2-Lite, it is more reasonable to primarily compare it with our own baseline, PureMoE.
>
> **Q2:**
> Since our method requires an MoE model as the base, it cannot use a dense model. At the time, there was no suitable MoE base model (with small parameter size and strong performance), so we selected DeepSeek-V2-Lite. In the future, MoE models with stronger performance can be used for further validation.
>
> **Q3:**
> This is more a hybrid workaround rather than a principled solution for the speech and text interference in parallel speech-text modeling？I'm a bit confused.
>
> **Q4:**
> The WER metric for speech generation performance can be referred to in Table 1.
>
> **Q5:**
> Since the model only gains general speech dialogue capability after the final training stage, it is difficult to accurately measure the model's true metrics during intermediate stages. Therefore, a proper analysis cannot be conducted effectively.
>
> **Q6:**
> Thank you for pointing out the citation format issue.

---

> ### Author Response · Authors · 2025-11-28
> **UTMOS and ASR-WER**
>
> For a comprehensive comparison, we evaluated the UTMOS for speech generation quality and the ASR-WER for both the generated speech and its corresponding generated text under the S2S setting, in order to verify whether the content of the generated speech matches the generated text.
> | model | UTMOS&uarr;  | ASR-WER&darr;|
> | :--- | :--- | :--- |
> |SpeechGPT| 3.86 | 66.57|
> | Mini-Omni | 3.17 | 25.28 |
> | LLaMA-Omni | 3.92 | 9.18 |
> | Moshi | 3.90 | 7.95 |
> | GLM-4-Voice | 4.45 | 5.74 |
> | DeepOmni | 4.23 | 3.43 |
>
> Experimental results indicate that the speech generated by DeepOmni achieves a commendable level of both audio quality and accuracy.
>
> The experimental results of replacing the base model with a stronger version, Qwen3-30B-A3B, can be found in the response to Reviewer YCDH.
>
> The comparative results for models such as Ming-lite-omni may be reported upon completion of the experiments.

---

> > ### Author Response · Authors · 2025-12-01
> > **Update Experiments**
> >
> > We tested the models you requested for comparison, such as Qwen3-Omni and Ming-Lite-Omni, under the Speech -> Text condition.
> >
> > | Model | LLaMA Question&uarr; | Web Question&uarr; | TriviaQA&uarr; | E2E Type |  Training Stage | Text Tokens | Text Conversations Num |
> > | :--- | :--- | :--- | :--- |:--- | :--- | :--- |:--- |
> > | DeepOmni(DeepSeek-V2-Lite-A2.4B) | 65.0 | 34.4 | 41.2 | Native |  SFT | 264M | 1.5M |
> > | Step-Audio2-Mini-7B | 69.0 | 42.4 | 50.3 | Native |  Pretrain + SFT | 400B | - |
> > | Kimi-Audio-7B |  75.3 | **62.8** | 57.1 | Modular | Pretrain + SFT | - | 12.7M |
> > | Qwen3-Omni | **85.7** | 57.9 | **77.1** | Modular | Pretrain + SFT | - | All text data of Qwen3-30B-A3B-Instruct-2507 |
> > |Ming-lite-Omni| 79.3 | 51.6 | 65.5 | Modular | Pretrain + SFT | | ~500M |
> > | DeepOmni(Qwen3-30B-A3B) | 73.0 | 48.3 | 52.1 | Native | SFT | 264M | 1.5M |
> >
> > As can be seen, since Qwen3-Omni and Ming-Lite-Omni are modular MLLMs and utilize a significantly larger scale of text training data compared to DeepOmni, their performance is indeed better. However, within the category of native MLLMs, we have achieved reasonably good results using a relatively small amount of data, undergoing only the SFT stage without pretraining. Additionally, Uni-MoE only supports text output and is not an end-to-end model for voice dialogue, therefore it is not included in the comparison.

---

### Author Response · Authors · 2025-12-03
**Author Final Remarks**

**Common Concern:**
Reviewers Vviv, yAcB, and YCDH all pointed out that the base model is not strong enough and suggested validating the performance on a stronger MoE-based base model, as well as comparing with stronger models such as Qwen-Omni, Step-Audio2, Kimi-Audio, and Ming-lite-Omni. We addressed this concern in our response to Reviewer YCDH, and YCDH acknowledged that the concern was resolved and raised the score accordingly.

**Individual Concern:** Reviewer Vviv suggested evaluating ASR-WER and UTMOS metrics under S2S experiments. We conducted the evaluations, and both content consistency and speech generation quality achieved satisfactory results. Vviv also recommended analyzing ablation experiments at each stage. However, only after the final training stage does the model acquire multimodal dialogue capabilities; the intermediate stages are not suitable for quantitative evaluation. Therefore, conducting ablation studies for each stage is challenging. Although we have not received a response from Vviv, we believe the main concerns have been addressed.

Reviewer mQes requested additional experimental comparisons on paralinguistic information. We supplemented the experiments with emotion-aware responses.

Reviewer yAcB argued that modular MLLMs also possess end-to-end differentiability—for instance, by utilizing multiple LLM hidden states to capture various paralinguistic cues—and questioned the necessity of native end-to-end modeling. We acknowledge this approach, but paralinguistic information in speech is highly diverse. Using multiple LLM hidden states to capture such information can be cumbersome. We aim for an end-to-end model that natively supports the understanding and expression of paralinguistic information, allowing the LLM to directly output such information without requiring the LLM hidden states to be processed by a speech LLM decoder. Hence, exploring native end-to-end modeling is necessary.

Reviewer YCDH raised concerns that batch parallel decoding increases computational overhead and suggested comparing it with a native MLLM that does not use batch parallel decoding. We provided a detailed explanation of the method's principles and resolved the reviewer's concern.

---

### Meta-Review · Area_Chair_renu · 2026-01-04

**Summary:**

This work presents a technique for improving multi-modal (speech and text) LLMs, using expert selection, followed by single-modality and cross-modality training stages. The resulting system results in less degradation in quality compared to other approaches.

Main issues raised by reviewers:
1. Weak baselines + missing important results across tables (like Qwen2.5-Omni, Kimi-Audio and Step-Audio2-Mini).
2. The use of a weak backbone to build the model, which is already lagging compared to other approaches.
3. Missing additional analysis of model components, other probing analysis / visualizations.
4. Missing emotion aware results.
5. Unsubstantiated claims about limitations of modular MLLM.
6. Concerns about clarity of presentation.

**Reviewer Concerns:**

1. The authors argue that models like Qwen2.5-Omni is not native end-to-end MLLM. But unless quality improves over modular MLLM, it is unclear why one would choose a native end-to-end. The authors shared additional results in the rebuttal, and also added results to the paper.
2. The authors mention that the backbone is already weak, so it is hard to improve. But again, the reviewers comments about not improving over a strong baseline remains.
3. The authors say that intermediate results cannot be computed, but it’s not clear why. Did not address this comment in detail.
4. The authors added this to rebuttal, but there’s very little context or explanation. So it’s hard to interpret.
5. The authors provide a handwavy response in their rebuttal.
6. The authors provided clarification, but unclear if all of this was incorporated into the paper.

**Reviewer Scores:**

Reviewer Vviv (Wen Wang): 4 -> 4.

Reviewer mQes (Jionghao Han): 6 -> 6.

Reviewer yAcB (Siddhant Arora): 4 -> 4.

Reviewer YCDH (Qian Chen): 4 -> 6.

---

### Decision · Program_Chairs · 2026-01-26

Reject